# Cerebellar Cells Self-Assemble into Functional Organoids on Synthetic, Chemically Crosslinked ECM-Mimicking Peptide Hydrogels

**DOI:** 10.3390/biom10050754

**Published:** 2020-05-12

**Authors:** Zbigniev Balion, Vytautas Cėpla, Nataša Svirskiene, Gytis Svirskis, Kristina Druceikaitė, Hermanas Inokaitis, Justina Rusteikaitė, Ignas Masilionis, Gintarė Stankevičienė, Tadas Jelinskas, Artūras Ulčinas, Ayan Samanta, Ramūnas Valiokas, Aistė Jekabsone

**Affiliations:** 1Institute of Pharmaceutical Technologies, Lithuanian University of Health Sciences, Sukilėlių ave. 13, LT-50162 Kaunas, Lithuania; zbigniev.balion@lsmuni.lt (Z.B.); justina.rusteikaite@lsmu.lt (J.R.); 2Neuroscience Institute, Lithuanian University of Health Sciences, Eivenių str. 4, LT-50161 Kaunas, Lithuania; natasa.svirskiene@lsmuni.lt (N.S.); gytis.svirskis@lsmuni.lt (G.S.); 3Ferentis UAB, Savanorių 231, LT-02300 Vilnius, Lithuania; vytautas@ferentis.eu (V.C.); kristina.druceikaite@ri.se (K.D.); masilioi@mskcc.org (I.M.); gintare@ferentis.eu (G.S.); tadas@ferentis.eu (T.J.); valiokas@ftmc.lt (R.V.); 4Department of Nanoengineering, Center for Physical Sciences and Technology, Savanorių 231, LT-02300 Vilnius, Lithuania; ulcinas@ftmc.lt; 5Institute of Anatomy, Lithuanian University of Health Sciences, Mickeviciaus 9, LT-43074 Kaunas, Lithuania; hermanas.inokaitis@lsmuni.lt; 6Polymer Chemistry, Department of Chemistry - Ångström Laboratory, Uppsala University, Box 538, 75121 Uppsala, Sweden; ayan.samanta@kemi.uu.se

**Keywords:** collagen-like peptide, collagen mimetic peptide, RGD, hydrogel, neurons, astrocytes, microglia, Ca^2+^ oscillations, tissue engineering

## Abstract

Hydrogel-supported neural cell cultures are more in vivo-relevant compared to monolayers formed on glass or plastic substrates. However, there is a lack of synthetic microenvironment available for obtaining standardized and easily reproducible cultures characterized by tissue-mimicking cell composition, cell–cell interactions, and functional networks. Synthetic peptides representing the biological properties of the extracellular matrix (ECM) proteins have been reported to promote the adhesion-driven differentiation and functional maturation of neural cells. Thus, such peptides can serve as building blocks for engineering a standardized, all-synthetic environment. In this study, we have compared the effect of two chemically crosslinked hydrogel compositions on primary cerebellar cells: collagen-like peptide (CLP), and CLP with an integrin-binding motif arginine-glycine-aspartate (CLP-RGD), both conjugated to polyethylene glycol molecular templates (PEG-CLP and PEG-CLP-RGD, respectively) and fabricated as self-supporting membranes. Both compositions promoted a spontaneous organization of primary cerebellar cells into tissue-like clusters with fast-rising Ca^2+^ signals in soma, reflecting action potential generation. Notably, neurons on PEG-CLP-RGD had more neurites and better synaptic efficiency compared to PEG-CLP. For comparison, poly-L-lysine-coated glass and plastic surfaces did not induce formation of such spontaneously active networks. Additionally, contrary to the hydrogel membranes, glass substrates functionalized with PEG-CLP and PEG-CLP-RGD did not sufficiently support cell attachment and, subsequently, did not promote functional cluster formation. These results indicate that not only chemical composition but also the hydrogel structure and viscoelasticity are essential for bioactive signaling. The synthetic strategy based on ECM-mimicking, multifunctional blocks in registry with chemical crosslinking for obtaining tissue-like mechanical properties is promising for the development of fast and well standardized functional in vitro neural models and new regenerative therapies.

## 1. Introduction

Numerous studies have indicated the limited capacity of the commonly used planar plastic and glass substrates to ensure physiologically relevant responses in cell cultures. It is well-known that the physicochemical properties of these materials are essentially different from those of the living tissues [1,2,3,4]. Therefore, there is a rapidly increasing need for establishing realistic tissue-like cell culture systems by employing both scaffold-free and scaffold-based approaches [1,2,5,6]. Hydrogels are among the favorable scaffolds because of their tissue-resembling properties such as a fibrous structure and softness, high content of bound water, and extracellular matrix (ECM) mimicking biochemistry [7,8,9,10]. However, there is still a shortage of hydrogel scaffold-based cell culture models that could be standardized, easily manipulated, and compatible with specific media or with high throughput workflows [11]. In regard to the brain cells, the challenge lies in providing a robust and repeatable assay, ensuring tissue-like cell composition and interaction and the efficient maturation of the functional network characterized by spontaneous neuronal firing activity [12]. The ECM components responsible for the cell attachment via focal adhesions are important players in cell differentiation and functional tissue formation [13]. Collagen and fibronectin have changing expression patterns in the developing cerebellum to control cell differentiation and migration [14,15]. Recently, different peptide fragments of collagen (CLPs) and the integrin-recognizing fragment of fibronectin (RGD) have been reported to mimic the effect of the entire protein molecules by enhancing neuronal adhesion, neurite outgrowth, and branching [16,17,18]. Such functional peptide building blocks are attractive tools for tissue engineering because they are relatively easy to synthesize and can be integrated into more complex synthetic materials. However, not only the presence of bioactive ECM-mimicking peptides but also other cues, such as the elastomechanical properties of the cellular environment, are important. For example, the stiffness plays a significant role in mediating focal adhesion formation by neural cells via mechanosensitive cellular membrane-embedded receptors such as integrins and integrin-regulating proteins [19].

The present study aimed to explore synthetic ECM-mimicking hydrogels based on CLP and CLP-RGD blocks conjugated to a poly(ethylene glycol) template (PEG-CLP and PEG-CLP-RGD, respectively), as well as to investigate how these artificial, all-synthetic substrates affect primary cerebellar cell organization, composition, motility and function. Recently, chimeric peptides have emerged as a promising synthetic biological strategy for medical applications [20], and we have already reported the application of PEG-CLP-RGD hydrogels for cancer cell models ([21], in revision). Additionally, we aimed to elucidate the role of the viscoelasticity of the ECM-mimicking substrate. Our study reveals a completely different but closer to the in vivo-like nature of the primary cerebellar assemblies obtained on the ECM-mimetic hydrogels as compared to the poly-L-lysine-coated laboratory plastic and glass substrates that are commonly used for neuronal-glial cultures.

## 2. Materials and Methods

### 2.1. Fabrication and Characterization of the Hydrogel Membranes

Unless otherwise stated, all chemicals were purchased from Sigma-Aldrich.

Peptide synthesis, the conjugation of the peptides to PEG, and the synthesis of hydrogels were carried out following the protocols described elsewhere [21,22]. The peptides CLP (Cys-Gly-(Pro-Lys-Gly)_4_(Pro-Hyp-Gly)_4_(Asp-Hyp-Gly)_4_) and CLP-RGD (-Arg-Gly-Asp-Ser-Pro-Gly) were synthesized by UAB Ferentis (Vilnius, Lithuania). For the conjugation of the peptides to PEG, 40 kDa 8-arm PEG-maleimide (hexa-glycerol core, Creative PEGWorks, NC, USA or JenKem, TX, USA) in dimethyl sulfoxide (DMSO) was mixed with an aqueous solution of CLP or CLP-RGD peptide at a molar ratio of PEG-maleimide:peptide 1:2.5 at room temperature. After 2 days of continuous stirring, a homogeneous solution of the peptide-PEG conjugate was dialyzed for 2 days using 12–14 kD MW cut-off tubing (Spectrum Laboratories, Inc., Rancho Dominguez, CA, USA), lyophilized, and stored at 4 °C until further use. 

For hydrogel synthesis, 12% (*w/w*) PEG-CLP and 12% (*w/w*) PEG-CLP-RGD aqueous stock solutions were prepared. Then, 500 mg of the stock solutions were taken into a T-piece 2 mL glass syringe mixing system [23]. The pH values of the solutions were adjusted to 4.5 by the addition of 2 M sodium hydroxide. After this step, calculated volumes of N-hydroxysuccinimide (NHS) and N-(3-dimethylaminopropyl)-N’-ethylcarbodiimide hydrochloride (EDC) were added to the mixing system as 10% solutions in 0.625 M 2-(N-morpholino)ethanesulfonic acid buffer (MES). The molar equivalent for peptide-NH_2_:NHS:EDC was 1:1:1. All the reagents were thoroughly mixed, and then the hydrogel was cast and kept between two glass slides with a 500 µm-thick spacer to control the thickness of the resulting membrane. It was left for curing for 16 h in a humidified chamber. The composition of peptides and the conjugation of CLP or CLP-RGD with 8-arm PEG-maleimide were characterized using ^1^H NMR on a Bruker Ascend 400 MHz spectrometer at room temperature. Briefly, 2% solutions of CLP, CLP-RGD, PEG-CLP, and PEG-CLP-RGD were prepared in D_2_O (Deuterium oxide). The resonance of deuterated solvent (D_2_O, δ = 4.79) was used as the internal standard.

### 2.2. Glass Functionalization by ECM Peptides

As model solid substrates, 10 mm diameter and 150 ± 10 µm thickness glass slides and 10 × 10 mm^2^ silicon substrates were employed. The latter substrates were cut from a 4′ silicon wafer (Topsil, Denmark). The substrates first were rinsed in ethanol and dried in a stream of N_2_ gas. Then, they were washed using the so-called SC-1 cleaning procedure: in a 5:1:1 mixture of ultrapure water, 30% H_2_O_2_, and 25% NH_4_OH solutions at 85 °C temperature for 10 min. Subsequently, they were treated in a plasma dry cleaner at the 20 W power (Femto, Diener Electronic GmbH, Ebhausen, Germany) for 3 min. After the plasma treatment, the glass substrates were silanized with (3-aminopropyl)trimethoxysilane (APTMS) (Thermo Fisher Scientific) [24]. Prior to the PEG-CLP and PEG-CLP-RGD coating process, a drop of 5% *w*/*w* glutaraldehyde solution in a 0.1M PB pH = 8.0 buffer was applied onto a surface for 20 min to convert the amine groups into aldehydes. After this step, the samples were rinsed with water and dried in the N_2_ gas. A solution of 2% PEG-CLP or 2% PEG-CLP-RGD in a PB buffer (0.1 M pH = 5.7) was applied onto the glass slides functionalized with the aldehyde groups and kept for 40 min at 37 °C temperature. After incubation with the respective peptide-PEG conjugate solutions, the samples were rinsed in water and dried in the N_2_ gas stream. The samples were stored dry in 4–8 °C until further use. The functionalized silicon substrates were investigated by using both imagining ellipsometry and atomic force microscopy (AFM). In contrast, the glass substrates were analyzed solely by AFM (see the respective Methods section below and the SI file for details). In total, 9 samples for each peptide-PEG conjugate were prepared and characterized.

### 2.3. Spectral Characterization of the Peptide Assembly

The three-dimensional structure of the peptides and the respective PEG-peptide conjugates was estimated using a J-815 circular dichroism (CD) spectropolarimeter (Jasco, MD, USA) equipped with a Peltier temperature-control system. Briefly, 1% (*w/w*) sample solutions in Milli-Q water with an adjusted pH = 5 were measured in a quartz cuvette with a 1 mm path length. Three spectra were collected and averaged for each type of peptide/conjugates from 190 to 250 nm at 25 °C at the 50 nm/min scanning speed.

### 2.4. Quantitative Characterization of the Elastomechanical Properties of Hydrogels

The mechanics of the fabricated hydrogel membranes was analyzed by measuring the two elastic parameters: shear storage modulus G’ and Young modulus E*. For G’ determination, oscillatory rheology was performed on a DHR-2 discovery hybrid rheometer (TA instruments, Sweden). All frequency sweep experiments were performed using 8 mm diameter stainless steel parallel plate geometry at 25 °C under an axial load of 150 mN. Circular discs of an 8 mm diameter were cut out from 0.5 mm-thick hydrogel membranes using a regular biopsy punch. Hydrogel samples were kept in 10 mM PBS. All experiments were performed using a shear strain amplitude of 0.27% at a frequency range of 0.02–50 Hz. To enable direct comparison with the Young modulus values obtained at a specific loading rate, storage modulus values obtained at 10 Hz were selected for presentation.

The AFM nanoindentation technique was employed to measure the elastic (Young) modulus E* values of the hydrogel samples [25,26]. The nanoindentation probes were prepared as follows. The spring constant of the tipless AFM cantilevers (NSC36-B, MicroMasch) was calibrated before the attachment of the microsphere particles using the thermal noise method [27] and was found to be 4.0 ± 0.6 N/m. The probes from one fabrication batch, which showed a low variation of the spring constant, were used. To ensure a controlled geometry of the contact, silicon dioxide microspheres (Microparticles GMbH) with a manufacturer-reported diameter of 6.65 ± 0.28 μm were attached to the AFM cantilevers using UV-curable glue (NOA68, Norland) while taking care to control the amount of glue on the cantilever. The attachment of the microspheres and their diameter was inspected by optical microscopy (x50 lens, BX51, Olympus, Japan), yielding the expected diameter of 6.65 ± 0.15 μm.

Cutouts of the hydrogel membranes with a typical width of 3 mm and a length of 5 mm were attached to the (3-aminopropyl)trimethoxysilane-treated glass substrates cut from regular carrier slides via glutaraldehyde chemistry (see above). In a typical experiment, one glass substrate had 2–5 hydrogel pieces mounted.

The elastic modulus measurement procedure was carried out as follows. The hydrogel sample was placed in a small Petri dish filled with PBS (pH = 7.4) and mounted onto the stage of an inverted optical microscope (IX73, Olympus, Japan), which served as a base for the AFM instrument (Nanowizard 3, JPK, Germany). The nanoindentation probe was approached to the hard glass surface, and the detection sensitivity factor was determined from the force-displacement curve. Typical detection sensitivity values were 10–20 nm/V. Then, the probe was repositioned over the hydrogel sample, and 1024 force curves were collected over the 50 × 50 µm area using the Quantitative Imaging™ (QI, JPK) mode with a force setpoint of 120 nN and a loading rate of 50 µm/s. For each sample analyzed, this procedure was repeated at 4 different locations.

The obtained curves were analyzed, and the values of the elastic modulus of the hydrogel samples extracted by fitting the approach part of the force-displacement curve with the Hertz sphere-on-plane elastic contact model using the JPK data processing software (version spm-5.0.84). The elastic modulus values were calculated from measurements of 4–12 different samples for every hydrogel formulation.

To compare the stiffness of the fabricated hydrogels, we estimated the compressive modulus (i.e., Young modulus) from the storage modulus at 10 Hz, as described in [28]. This was done by using the following equation: E = 2G’ (1 + ν)(1)
where E is the compressive modulus, G’ is the storage modulus, and ν is the Poisson’s ratio; for hydrogels, ν = 0.5.

### 2.5. Surface Topography and Thickness Measurements of Membranes and Coatings

Surface topography images of the PEG-CLP and PEG-CLP-RGD hydrogel membranes and their respective coatings on glass were acquired using the same AFM instrument (NanoWizard 3, JPK) with the MLCT-A (Bruker) and NSC35-C (MicroMasch) AFM probes in a PBS pH = 7.4 buffer solution. Imaging was done using the QI mode at the 0.4–5 nN force setpoint; different regions of each sample were scanned. The scan size ranged from 50 × 50 to 1 × 1 µm^2^ at the 256 × 256 pixel resolution. Three samples from three different fabrication batches were analyzed for each hydrogel type.

The thickness of the organic layers on the silicon substrates was measured with an imaging null ellipsometer (Nanofilm_ep3, Accurion GmbH, Goettingen, Germany), using a laser emitting at the 658 nm wavelength and a 10× objective. The measurements were carried out at the 70° angle of incidence unless otherwise stated. Thickness modeling was performed with the Nanofilm_ep4 Model software (Accurion GmbH, Goettingen, Germany). The actual substrate layer thicknesses were fitted from the angle of incidence spectrum, measured on a substrate freshly cleaned by the SCI-1 procedure (see above). The coatings of APTMS, GA, and the PEG-peptide conjugates analyzed in air were modeled as organic layers with a refractive index of n = 1.5 and k = 0. The measurements were performed after each step of coating. For statistics, 9 samples of the conjugates bound to glass were measured.

### 2.6. Preparation of Neuronal-Glial Cell Culture and Viability Assessment

All experimental procedures were performed according to the Guide for the Care and Use of Laboratory Animals. The rats were maintained at Lithuanian University of Health Sciences animal house in agreement with the Guide for the Care and Use of Laboratory Rats. Cerebella were isolated from euthanized postnatal 5–7-day-old Wistar rats and minced into small cubes (ca. 0.5 mm^3^). The minced tissue was transferred into a Falcon tube containing 7 mL of a Versene solution (1:5000; Gibco, Thermo Fisher Scientific) and incubated at 37 °C for 5 min. The tissue was further triturated with a Pasteur pipette to obtain a single-cell suspension. The incubation and trituration procedures were repeated until any undissociated tissue remains became invisible. The single-cell suspension was centrifuged at 270 g for 5 min and resuspended in a DMEM medium with Glutamax (Thermo Fisher Scientific) supplemented with 5% horse serum, 5% fetal calf serum, 38 mM glucose, 25 mM KCl, and antibiotic-antimycotic (Thermo Fisher Scientific). The cells were plated at a density of 0.25 × 10^6^ cells/cm^2^ in 96-well plates (VWR) on disc-shaped hydrogel membranes or on the plate bottom coated with 0.0001% poly-L-lysine. The cells were kept at 37 °C in a humidified incubator containing 5% CO_2_. On in vitro days 4 and 7, the viability of the cultures was assessed by double nuclear staining with fluorescent dyes Hoechst33342 (6 µg/mL) and propidium iodide (3 µg/mL) for 10 min and imaging in a fluorescent microscope Olympus IX71 (Olympus Corporation, Tokyo, Japan) equipped with a ×20 objective. The images were taken by a 01-Exi-AQA-R-F-M-14-C camera (QImaging, Surrey, Canada). The image analysis was performed by the ImageJ software. 

### 2.7. Preparation of Pure Microglial Cell Culture

Primary mixed astrocyte and microglial cultures were used for pure microglial culture preparation. First, mixed glial cultures were prepared from the cerebral cortices of 5–7-day-old Wistar rats. After the dissection of the cerebral hemispheres, the meninges were removed, and the tissue was dissociated in a solution of EBSS containing 0.3% BSA, 103.2 Kunitz units/mL DNase I, and 3800 BAEE units/mL trypsin. The cells were plated at 2 × 105 cells/cm^2^ in 75 cm^2^ flasks coated with 0.0001% poly-L-lysine. The cultures were maintained in DMEM supplemented with 10% fetal calf serum and 1 mg/mL antibiotic-antimycotic (Gibco). The cells were kept at 37 °C in a humidified atmosphere of 5% CO_2_ and 95% air. After the mixed glial cultures reached confluence (on the 6th–8th days in vitro), the 15 microglial cells were isolated by shaking and tapping the flasks. The medium from the mixed glial cultures, containing separated microglial cells, was removed and centrifuged at 135 × *g* for 5 min. The supernatant was discarded, and the cells were resuspended in DMEM with the same supplements as those of the mixed glial cultures and plated at a density of 2 × 105 cells/cm^2^ in the uncoated flat bottom 96 well plates (VWR) with or without hydrogel inserts (membranes). The cells were kept at 37 °C in a humidified incubator containing 5% CO_2_ and were later used for the proliferation assessment.

### 2.8. Evaluation of Cell Number, Composition, and Neuritogenesis

All nuclei were stained with Hoechst33342 (6 µg/mL, 15 min at 37 °C). Neurons were identified by immunostaining for microtubule-associated protein 2 (MAP2) and astrocytes by immunostaining for glial fibrillary acidic protein (GFAP). The cultures were fixed in 4% paraformaldehyde in PBS for 20 min, permeabilized in 0.3% Triton X-100 in PBS, blocked with 10% BSA in PBS, and incubated for 1h with primary antibodies: 1 µg/mL of rabbit polyclonal anti-MAP2 (Abcam) and 4 µg/mL of mouse monoclonal anti-GFAP (Thermo Fisher Scientific), as well as 30 min with the secondary antibody AlexaFluor^®^555 conjugated goat anti-mouse IgG (Invitrogen) and AlexaFluor^®^647 conjugated chicken anti-rabbit IgG (Thermo Fisher Scientific), both diluted in PBS 1:200. Fixed microglial cells in the cultures were detected by isolectin GS-IB_4_ from *Griffonia simplificolia*, Alexa Fluor^®^ 488 conjugate (10 ng/mL for 15 min at 37 °C, Molecular Probes). The cells in cultures were detected using laser scanning confocal microscopes: Zeiss Axio Observer LSM 700 (Carl Zeiss Microimaging Inc., Jena, Germany) and Olympus Fluoview FV1000 (Olympus Corporation, Tokyo, Japan). Each image displayed a maximum intensity projection of a stack of 5–12 z-sections of 3–6 different cultures. The total cell number on the hydrogel, glass, and plastic samples was evaluated by counting the nuclei in the 400 × 500 μm^2^ area of the confocal micrographs. For neurite evaluation, the anti-MAP2-positive area in micrographs was calculated by the ImageJ software and expressed as the percentage of the total image area per neuron.

### 2.9. Evaluation of Microglial Proliferation

Microglial proliferation in mixed neuronal-glial cultures was assessed by the daily cell counting of isolectin GS-IB_4_ positive cells in fluorescent images and in pure microglial cultures in phase-contrast images by the use of the inverted OlympusIX71 microscope (Olympus Corporation, Tokyo, Japan). Cells in the micrographs were counted by ImageJ software.

### 2.10. Cytokine Detection

The cell culture medium for cytokine detection was collected each day starting from the 1st day and finishing on the 6th day in vitro. The levels of pro-inflammatory cytokines in the neuronal-glial cell medium were detected by ELISA kits for rat TNF-α (Thermo Fisher Scientific) and IL-1β (Abcam) according to manufacturer’s protocol. A lipopolysaccharide (LPS)-activated cell culture medium was used as the positive control. The optical density in the samples was measured in a MultiskanFC plate reader (Thermo Fisher Scientific).

### 2.11. Neuronal Activity Monitoring by Ca^2+^-Sensitive Fluorescence

For Ca^2+^ oscillation measurement, the neuronal-glial cells on days 4 and 7 in culture were pre-loaded with the cell-permeable Ca^2+^-sensitive dye Oregon Green™ 488 BAPTA-1, AM (OGB-AM, Thermo Fisher Scientific). A solution of OGB-AM (5 µM) was added to the cell culture medium, and the cultures were placed to the incubator for 15 min. To wash out the dye that had not penetrated into the cells, the medium was replaced with fresh DMEM with all the supplements used for the mixed neuronal-glial cell cultures (as described in 2.6 above), and the cells were further incubated for 10 min. For Ca^2+^ signal registration, we applied the methods used in [29]. The LED light was limited to the recording site by a diaphragm. Images were acquired with the Solis software (Andor Technology Ltd., Belfast, UK) and stored on the disk for further analysis. Image analysis was performed with the Solis analysis software, as well as by employing the image analysis package ImageJ and custom written subroutines. The cultured cells stained with calcium-sensitive dye OGB-AM were registered in an area of 640 × 250 pixels. In every culture, 4–6 different regions were registered. The fluorescence images were acquired for 20 s with a rate of 30 Hz. In order to identify the cells with large spontaneous changes in calcium concentration, the image stacks were analyzed by custom-written Python subroutines by using the SciPy open-source software package. The fluorescence traces from single pixels were selected if the peaks were larger than 5% of the time-averaged fluorescence and the amplitude of the peak was 4 times larger than the smallest standard deviation in pixels of the corresponding image stack. Since the cells were imaged by a number of pixels, the detected signals were summed and averaged if they were all present in the neighboring pixels in the area smaller than 20 × 20 pixels. The areas with the detected signals in the image stacks were also inspected by using the ImageJ software package. The time course of the detected fluorescence signals was corrected for bleaching by the simple ratio method [30]. In this procedure, the decaying mean intensity of fluorescence in the region outside the detected signals was used to calculate the decay ratio for each image and to compensate for the bleaching in the time course of the signals. The time constant, τ, of the exponential decay for fluorescence signals was calculated by using the curve_fit procedure from the SciPy package, which uses the non-linear least-squares method to fit the first-order exponential decay. The average square error at each time moment was smaller than 0.05%.

### 2.12. Statistical Analysis

The statistical significance of the elastic modulus (E*) was evaluated by ANOVA (SPSS software) [F(1, 61) = 29.108, *p* = 0.000]. All quantitative data in the graphs are presented as means of 4–7 experiments and standard error. The graphs were made by and statistical significance was evaluated by the SigmaPlot v13 software by a one-way ANOVA Tukey test. The statistical analysis for the fluorescence data was conducted using the procedures from the SciPy package. The normality of data distribution was assessed by using D’Agostino and Pearson’s normality test. The statistical significance of the difference between the averages was assessed using Student’s t-test for normally distributed data. The surface roughness and layer thickness data are represented as mean values ± SDV.

## 3. Results

### 3.1. Chemical, Mechanical and Structural Properties of PEG-CLP and PEG-CLP-RGD Hydrogels

We synthesized the CLP and CLP-RGD peptides and confirmed their composition by ^1^H-NMR (Figure 1a). Note that arginine (δ = 3.19) and serine (δ = 3.83) peaks [31] in the NMR spectra of CLP-RGD are evident, and they confirmed a successful CLP sequence extension by the RGD block (Figure 1a). In the next step, we covalently attached the respective peptides to an eight-arm PEG template by the method described previously in [22]. Using CD spectroscopy, we assessed the ability of the peptide and peptide-PEG conjugates to form intramolecular assemblies (Figure 1b,c). For CLP, one could observe the same trend as the one previously reported by Islam et al. [22].

Hydrogels for the cell culture experiments were prepared from PEG-CLP ad PEG-CLP-RGD conjugates crosslinked with EDC/NHS, as schematically depicted in Figure 1d. The final PEG-peptide concentration in hydrogels cast from the 12% (*w/w*) stock solution was 8.5 ± 0.3% for both PEG-CLP and PEG-CLP-RGD. All the prepared hydrogel membranes were transparent and sufficiently mechanically robust to retain their shape and withstand handling in cell culture. For the quantitative determination of the elastomechanical characteristics, we carried out oscillatory rheology and AFM nanoindentation measurements. The first technique is used as a standard for the characterization of macroscopic polymer samples, whereas the second provides insights in the interfacial properties important in the context of mimicking the cellular microenvironment.

The results are summarized in Table 1. We chose the shear storage modulus G´ and the elastic (Young) modulus as the key parameters of the mechanical properties of the two hydrogel formulations analyzed. The shear storage modulus G´ characterizes the elastic response of the material to deformation in the direction parallel to the surface, whereas the stiffness characterizes the elastic response of the material to compressive deformation in the vertical direction. The values of G´ obtained by oscillatory rheology for PEG-CLP and for PEG-CLP-RGD were 26.7 ± 2.1 and 20.4 ± 5.5 kPa at 10 Hz, respectively. In turn, the compressive modulus E values calculated from G´ according to the Equation (1) were 80.1 ± 6.3 and 61.2 ± 16.5 kPa, respectively. For comparison, the compressive modulus E* values obtained from AFM nanoindentation for PEG-CLP and PEG-CLP-RGD were 144.5 ± 26.5 and 75.6 ± 19.0 kPa, respectively. Both the shear storage and elastic moduli showed the same tendency, with PEG-CLP having the higher stiffness than PEG-CLP-RGD.

Further on, we analyzed the native nanometer-scale structure of PEG-CLP and PEG-CLP-RGD hydrogel surfaces using AFM in PBS (pH = 7.4). This analysis revealed relatively smooth and densely packed hydrogel surfaces (see Figure 2a,b). The surface roughness values measured as root mean square (RMS) for PEG-CLP and PEG-CLP-RGD were 4.6 ± 2.8 and 2.7 ± 1.6 nm, respectively.

As controls for revealing the effects of the elastomechanical properties and three-dimensional arrangements of the peptides on the cell behavior, we also prepared glass substrates functionalized with the PEG-peptide conjugates. Specifically, we performed covalent coupling between the functional amine groups on the peptide and the aldehydes introduced on the surface. For process optimization, we first tested the surface chemical modifications of silicon wafer substrates with the native SiO_2_ layer. This allowed us to employ imaging ellipsometry as means for the careful monitoring of every step of the surface modification process. After the treatment by APTMS, the incremental thickness on the silicon surface was 0.30 ± 0.04 nm. After the additional treatment with a 5% glutaraldehyde solution, the thickness increased to 0.56 ± 0.03 nm, suggesting glutaraldehyde reaction with the surface amine groups. After the PEG-CLP solution incubation and washing procedure, the resulting total thickness of the formed layer was 4.23 ± 2.04 nm, whereas the incremental thickness due to the PEG-CLP attachment was 3.67 ± 2.04 nm. Subsequently, we employed the same protocol for the modification of the glass substrates for cell culture experiments. Since the ellipsometry technique cannot be used for the characterization of transparent glass substrates, we relied on the AFM analysis of the glass surface modified with the PEG-peptide conjugates in PBS (pH = 7.4). The AFM images revealed significant differences in the surface morphology of the conjugate coatings on the glass as compared to the surface morphology of the hydrogel membranes discussed above (Figure 2c,d; see also Appendix A, an AFM image of a GA-treated glass surface). In liquid, the surfaces exhibited inhomogeneous distribution of the material rather than smooth coatings. The observed wrinkle-like patterns seen for both coatings could have been a result of the non-homogeneous expansion of the surface-bound conjugate layer upon hydration [32]. From the point of view of the cellular attachment, it was important to verify whether the conjugates were covering the entire surface, preventing the undesired interaction with the underlying chemical groups and the SiO_2_ top layer itself. We detected no cracking or similar defects of the conjugate coatings, which would have led to the underlying glass surface being exposed. This could be further corroborated by the fact that the slope of the QI curves (which is indicative of the surface layer stiffness) recorded on the control (glass) surfaces was one order of magnitude higher than on the conjugate-coated substrates (data not shown).

### 3.2. Evaluation of Cerebellar Cell Cultures on PEG-CLP and PEG-CLP-RGD Hydrogels

Collagen and fibronectin are expressed in the developing cerebellum to control cell differentiation and migration [14,15,33]. The fibronectin fragment RGD mimics the effect of the entire protein by enhancing neuronal adhesion and neurite outgrowth [16,34,35]. In this study, the collagen structure mimicking peptide CLP alone and supplemented with the RGD motif were evaluated as the neural cell culture substrates for cells from the developing rat cerebella, taking into account the induced cellular composition, organization, neurite network formation capacity, and neuronal function. There were two distinct controls that we ran in parallel. A control to elucidate the influence of the viscoelasticity on cell behavior was done on glass coated with the PEG-peptide conjugates, and a control of conventional culturing of the cerebellar cells was made on laboratory glass and plastic covered with poly-L-lysine.

#### 3.2.1. Cerebellar Cell Culture Development and Organization

On the first day in vitro (DIV1), the isolated cerebellar cells formed irregular clusters on both PEG-CLP and PEG-CLP-RGD hydrogel membranes (Figure 3a). They completely attached to the poly-lysine-coated plastic or glass (Figure 3c,d). Clusters similar to those on hydrogels also formed on the glass modified with the PEG-CLP- and PEG-CLP-RGD conjugates; however, only a very small portion of the clusters attached to those coatings, and most of them were floating in the medium (Figure 3b). Going from DIV1 to DIV3, the cells on the PEG-CLP hydrogel formed spheroids of 45–185 µm in diameter with defined edges made of tightly packed granule neurons, and the structure remained visually similar until DIV7 (the last day of monitoring). On the PEG-CLP-RGD hydrogel, the cells then spread on the surface until DIV3 and started to reorganize, resulting in more defined spheroids with outgrowing fibers on DIV5. The size of the spheroids was similar to those on the PEG-CLP hydrogel (from 34 to 210 µm). On polylysine-coated plastic or glass, the cells were evenly spread and well attached on DIV1, and they remained evenly distributed during all 7 days of monitoring (Figure 3c,d).

#### 3.2.2. Cell Culture Composition and Neuritogenesis

On DIV7, the cells were fixed and stained for the neuronal, astrocyte, and microglial markers to identify the composition of the cultures. On the PEG-CLP hydrogels, the spheroids of tightly packed granule neuron bodies were surrounded by outspreading fibers composed of neurites alongside with adherent astrocytes (Figure 4a, upper images). Microglial cells were located around neurite-astrocyte fibers and contact neurites by their filopodia. We found a similar organization of the cells also on the hydrogels containing the RGD motif (Figure 4a, bottom panels). There were granule neuron body spheroids, a close location of astrocytes to neurons, and scattered branched microglial cells contacting neurites and/or astrocytes. However, the neurites in the cultures were not so compactly organized as on the PEG-CLP membrane.

There were almost no neuronal spheroids in the cultures grown on the PEG-CLP coated glass on DIV7 (Figure 4b, upper images). Both types of glial cells looked morphologically similar to those on the PEG-CLP hydrogel membrane. However, in contrast to hydrogels, there were no visible neurite-astrocyte connections in PEG-CLP glass-supported cultures. The spheroids formed on glass coated with the PEG-CLP-RGD conjugate were much more compact as compared to the hydrogel membranes made from the same material (Figure 4b, lower images). Additionally, the number of neurite outspreads was lower, neurite network less expressed, and the number of astrocyte cells decreased. Besides, the shape of the microglial cells was different compared to the cells on the hydrogel membranes. On the PEG-CLP-RGD coatings on glass, microglia had no filopodia but appeared either round or, on the contrary, very much attached to the surface and unusually flattened.

On poly-lysine coated glass and plastic, the cultures were extremely similar and characterized by the evenly distributed spreading of neurons, astrocytes, and microglia, though they lacked any specific organization pattern, such as spheroids, neurite-astrocyte co-localization, or a specific location of microglial cells (Figure 4c).

Total cell numbers were nearly twice as lower on the PEG-CLP hydrogels and by one third lower on PEG-CLP-RGD hydrogels compared to poly-L-lysine-coated glass and plastic cultures (Figure 5a). For comparison, there were only a few cells attached to the PEG-CLP-coated glass surface and slightly yet significantly higher number of cells on the PEG-CLP-RGD-coated glass.

The assessment of the neuronal, astrocyte, and microglial cell numbers revealed that there were significantly larger amounts of glia and, in particular, microglial cells on both hydrogel membranes and the respective conjugate-coated glass cultures compared to the cultures on poly-L-lysine on glass and plastic, respectively (Figure 5c).

The ability of the materials to stimulate and guide the neurite outgrowth might be important for neural tissue design and regeneration purposes. The hydrogel materials investigated in the study were also assessed for neurite network formation capacity. Neuritogenesis represented as the area covered by neurites per neuronal cell was significantly higher on hydrogels compared to cultures on plastic and glass, both covered by poly-L-lysine and PEG-peptide monolayers (Figure 5b). On the PEG-CLP-RGD hydrogel, neuritogenesis was extremely intensive, reaching 2% of the total cell culture area covered by neurites per one neuronal cell. Additionally, both PEG-CLP and PEG-CLP-RGD conjugates promoted significantly higher neuritogenesis when attached to the glass surface compared to poly-L-lysine.

#### 3.2.3. Microglial Proliferation and Inflammatory Cytokine Release

We estimated the microglial cell numbers in the cerebellar cell cultures on hydrogels. To find out whether microglia were directly stimulated for proliferation by the hydrogel material or if this was a result of altered intercellular signaling on the hydrogel membranes, we performed a daily assessment of the microglial cell number in pure microglial and mixed neuronal-glial cultures. The significant difference in the microglial number on both PEG-CLP and PEG-CLP-RGD hydrogels compared to the cultures on poly-L-lysine-coated plastic appeared from the third day in vitro and continued to increase until the end of monitoring on day 7 (Figure 6a). However, the hydrogels did not stimulate cell proliferation in pure microglial cultures during the same 7 day period of monitoring. The results indicated that microglial proliferation in mixed neuronal-glial cultures was more likely a result of cellular crosstalk on the hydrogel substrates (membranes) rather than a direct interaction of the hydrogel with microglia.

The increased numbers of glia in the rat cerebellum are typical for the first two postnatal weeks because of the intensive participation of the cells in neuronal network formation [16,36]. On the other hand, this might also have been a sign of an inflammatory response. To test whether this was the case in our hydrogel cultures, we assessed the TNF-α and IL-1β inflammatory cytokines released to the cell incubation medium. As increased microglial proliferation was observed only in the mixed neuronal-glial but not in the pure microglial cultures, we measured the amounts of the cytokines in the mixed culture medium by ELISA. An LPS-activated cell culture medium was used as a positive control. The amounts of TNF-α and IL-1β remained unchanged in the medium from the cultures growing on the PEG-CLP hydrogels during all 6 days of monitoring (Figure 6c,d). In the PEG-CLP-RGD culture medium, we found a slightly increased amount of TNF-α on days 1 and 6 after plating as compared to the PEG-CLP hydrogel cultures. We also observed elevated TNF-α in the medium from poly-L-lysine-coated plastic surface cultures on day 6 after plating. However, these levels were far from the peak generated after 6 h microglial stimulation with LPS. The PEG-CLP-RGD hydrogel and poly-L-lysine-plastic surface cultures also generated higher levels of IL-1β compared to the cells on the PEG-CLP hydrogels on DIV5 and DIV6 (Figure 6d), but these amounts were much lower than those that were generated by the LPS stimulation. Such results indicated that the observed higher microglial activity on hydrogels was not due to the inflammatory response but was most likely related to the selective clearance of neurons and synapses because it occurred in the native conditions, i.e., in the developing brain.

#### 3.2.4. Neuronal Synaptic Activity

In order to compare the functional states of neurons grown on the different substrates, we recorded fluorescence signals evoked by Ca^2+^ fluxes after neurons were stained with the membrane-permeable calcium-dependent fluorescence dye OGB-AM (Figure 7a,b). Spontaneous activity in cultured neurons is necessary for the formation of the synaptic contacts and overall maturation of neurons. The neurons acquired spontaneous activity depending on the surface substrate. Neurons on day 4 and 7 in vitro after explanting on poly-L-lysine coated tissue culture plastic had a very low probability of being spontaneously active—less than 10 cases in 10 cultures. On the other hand, a larger number of neurons (more than 50 cases in 5–7 cultures) had an activity of about 0.05–0.6 Hz on day 4 and 7 in vitro in cultures grown on the PEG-CLP or PEG-CLP-RGD hydrogel membranes. On day 7 in vitro, the recorded neurons displayed apparently synchronized activity (Figure 7c). In 50% of cases, two or more neurons showed synchronized signals: 16 cases in 33 recordings on PEG-CLP and 20 cases in 40 recordings on PEG-CLP-RGD. When three or more neurons had synchronized signals, seven such cases (21.1%) were recorded on PEG-CLP, and 15 (37.5%) cases were recorded on PEG-CLP-RGD. The cultures grown on glass coated with the PEG-CLP and PEG-CLP-RGD conjugates had no active cell clusters formed, most likely because of there not being enough cells attached to the surface. Thus, the PEG-CLP and PEG-CLP-RGD hydrogels supported the best functional maturation of cultured neurons of all the tested substrates. We further analyzed the properties of the signals from the cultures grown on these two types of hydrogel membranes substrates.

Neurons on day 4 in vitro in cultures grown on PEG-CLP hydrogels had the activity with a frequency equal to 0.30 ± 0.23 Hz (n = 74), which was significantly (*p* < 0.01) higher than in neurons on the PEG-CLP-RGD hydrogels at 0.20 ± 0.14 Hz (n = 60) (Figure 8a). However, on day 7, the difference between the frequencies of spontaneous signals was absent—they were equal to 0.27 ± 0.28 (n = 48) and 0.25 ± 0.21 Hz (n = 72) for neurons on the PEG-CLP and PEG-CLP-RGD hydrogels, respectively. Thus, the PEG-CLP hydrogel was a better substrate only for the initial growth period.

We also analyzed the strength of the spontaneous Ca^2+^ fluxes reflected in the relative amplitude δF/F of the fluorescence signals (Figure 8a). In neurons after 4 days in vitro on the PEG-CLP hydrogel membranes, the relative signal amplitude was equal to 6.15 ± 4.58% (n = 74), while the signals from the neurons on the PEG-CLP-RGD hydrogels had a significantly (*p* < 0.001) smaller amplitude at 3.08 ± 2.45% (n = 60). Thus, it seems that for the initial growth, the PEG-CLP hydrogels were more supportive, which corresponded to the result for the frequency of the spontaneous signals. However, after 7 days in vitro, the difference between the signal strength was abolished; the amplitudes were equal to 5.33 ± 2.57% (n = 48) and 5.31 ± 3.24% (n = 72) for neurons on the PEG-CLP and PEG-CLP-RGD hydrogels, respectively (Figure 8b). 

The functional state of the neuron depends on the effectiveness of intracellular calcium handling. After the influx of Ca^2+^ due to excitation, a healthy neuron reduces calcium concentration to the resting level in the most efficient way, which is reflected in the rate of the decay of fluorescence signals. We characterized the decay of calcium concentration by fitting the decay of the fluorescence trace with a single exponential function and by obtaining its characteristic time constant, τ. Neurons, after 4 days in vitro on the PEG-CLP or PEG-CLP-RGD hydrogels, had similar values of τ equal to 1.46 ± 0.68 (n = 46) and 1.38 ± 0.65 s (n = 24), respectively. The decay of the signals in neurons on day 4 in vitro on the poly-L-lysine-coated tissue culture plastic was much slower; τ was equal to 2.26 ± 1.28 s (n = 8). The difference was not significant due to a very low number of active neurons on poly-L-lysine-coated plastic, as described above. After 7 days in vitro, the values of τ became different: the neurons on the PEG-CLP-RGD hydrogels had a smaller τ equal to 1.22 ± 0.71 s (n = 51) compared to the value of 1.72 ± 1.25 s (n = 40, *p* < 0.05) from the neurons on the PEG-CLP (Figure 8c). The much faster decay of intracellular Ca^2+^ in neurons on the PEG-CLP-RGD hydrogels showed much better conditions for their development for the period longer than 4 days in vitro.

## 4. Discussion

The materials synthesized as the mimetics of the natural ECM that were analyzed in this study differ from most of the hydrogels reported in the literature [37] and/or those available commercially. While the peptide fragments of collagen have been used as substrates for cell cultures [38,39], in our PEG-peptide hydrogels, the native-like assemblies of the peptides are facilitated by the covalent attachment to an eight-armed PEG-maleimide. The overall architecture is further locked by chemical crosslinking, as described in [22]. Besides that, such a synthetic strategy is advantageous and flexible in terms of (bio)chemical variation by introducing and combining different active ECM elements. For example, in our case, we employed the CLP peptide sequence and the C-terminus extended with a different functionality—the cell adhesion peptide RGD—as the building blocks of the hydrogel [21]. 

The circular dichroism analysis of CLP, CLP-RGD, and their respective PEG conjugates showed the triple helix signature at a wavelength of 225 nm (Figure 1b). Conjugation to eight-arm-PEG further promoted the self-assembly for PEG-CLP, as was observed before [22]. However, we found that for CLP-RGD, this signal was significantly weaker compared to the unmodified CLP block, thus suggesting that the introduction of RGD partially hindered the formation of the triple helical structure. 

From the elastomechanical analysis (Table 1), one can judge that both shear storage and elastic moduli showed the same tendency, with PEG-CLP having the higher stiffness than PEG-CLP-RGD. The decrease in stiffness for PEG-CLP-RGD hydrogel compared to PEG-CLP could be partly attributed to the reduced helicity of these conjugates compared to PEG-CLP. Nevertheless, the moduli values obtained for both synthesized hydrogels were within the range of those measured for brain tissues [40] and at least by four orders of magnitude lower than those of polystyrene [41] commonly used as a substrate for in vitro cell cultures. The calculated elastic modulus (E’) from the shear modulus (G’) data yielded similar values to the elastic modulus (E*) obtained by the AFM nanoindentation-based technique (E*). The AFM nanoindentation assay measured deformation force in submicrometer-sized surface regions, and it was limited by the low depth of indentation and the actual dimensions (6.65 ± 0.28 µm diameter) of the silicon dioxide microsphere used as the indenter. As such, this parameter is arguably a better indicator of the mechanical properties of the substrate as sensed by a living cell. On the other hand, dynamic oscillatory rheology revealed the mechanical properties that are associated more with the macroscopic properties of the bulk of the material. These data are more relevant for technical operations with the hydrogel membranes such as handling during the cell culture, post-culture analysis, medical techniques like trepanning, and suturing.

The RGD peptide promotes cellular adhesion and acts as a neuritogenic fragment [16,31,32]. However, the different stiffnesses of the PEG-CLP and PEG-CLP-RGD hydrogels may have also affected the cell organization on the hydrogels surface. As an attempt to separate the biochemical and the mechanical cues, we performed a comparison of the cell culture on both the elastic hydrogel substrates and the more rigid coating of the PEG-peptide conjugates on a solid surface. We expected that in the latter case, the interaction between the peptides and the cellular receptors would be more sterically hindered. Indeed, the attachment of cells to the PEG-CLP and PEG-CLP-RGD coatings on glass was poor compared to the attachment to the respective peptide hydrogels. One has to bear in mind that the surface concentration of the peptide moieties available for interaction with the cells could be lower on the glass substrates as compared to the hydrogels. Upon making the coating, the PEG-peptide conjugates were chemically coupled to the glutaraldehyde-modified glass substrates via the amine groups present on the peptides. Additionally, the peptides tethered to the glass surface had less steric freedom to form the triple-helical assemblies, and this could also negatively affect molecular recognition. Nevertheless, the number of cells attached to the PEG-CLP-RGD coating was two times higher than that on the PEG-CLP coating. This was in favor of a receptor-mediated interaction between the cells and the peptide molecules on the surface. However, for revealing a detailed picture of these interactions, one should design a separate study including synthesis of model surfaces with a controlled and tunable presentation of the peptides (also scrambled sequences) and the performing of molecular characterization. 

We also observed the same effect of the RGD motif promoting the cell adhesion upon comparing the cultures on the PEG-CLP-RGD and PEG-CLP hydrogels. The RGD motif promoted cellular attachment despite at the same time restricting the formation of the helical assemblies of the peptides (see above). Thus, the above observations correlate with the previous studies, which demonstrated that RGD-functionalized membranes promote neural cell adhesion and aggregation [42,43].

Interestingly, upon seeding the cells on the PEG-CLP-RGD hydrogel, they initially spread, and after several days, they started clustering. Whereas on the PEG-CLP hydrogel, the cells already formed compact spherical clusters during the first day after seeding. These cellular assemblies remained present during the whole monitoring period. Both collagen and fibronectin are known to have distinct roles in the formation of cerebellar shaping during development by cell differentiation and migration control [14,15,30]. Thus, we mimicked this effect by employing the chimeric CLP-RGD peptide rather than formulating hydrogels from individual motifs. The introduction of RGD made the cultures more “relaxed,” not so tightly packed, and potentially more prone to migration, as one might guess from the observed more scattered glial cell distribution.

The highest number of cells was on the poly-L-lysine-coated plastic and glass surfaces. Moreover, the neuronal cells highly predominated in these cultures. Not surprisingly, these coatings are very common for neural cell cultures in the laboratories. Giving such a high yield of firmly attached neuronal cells, poly-L-lysine made the neural cell cultures easy manageable. On the other hand, the cells in such cultures were immobilized and did not show any indication for organization or communication. Contrary to this behavior, on the hydrogel membranes, the cerebellar granule cells were gathered to compact clusters resembling the cerebellar granule layer [15]. Additionally, the astrocytes were distributed alongside the neurite fibers that connected the clusters into a distinct network. Such a cellular organization suggests there was ongoing functional communication between the neuronal and glial cells in these complex cultures.

The cerebellar cells grown on hydrogels and PEG-peptide-coated surfaces had higher glial to neuron ratios and elevated numbers of microglia. Perez-Pouchoulen and colleagues found that microglial percentages in developing rat cerebellum vary from about 20 to 60 during the postnatal 21 day period [44]. Microglial numbers on both the PEG-CLP and PEG-CLP-RGD hydrogels and glass surfaces were within this range, indicating the materials support in vivo-relevant microglial/non-microglial cell ratios. On the other hand, the total cell numbers on PEG-peptide-coated glass surfaces were more than five times lower than on hydrogels and did not support functional cell assemblies. On poly-L-lysine-coated plastic or glass, microglial percentages were far below those found in real cerebellar tissue. In rat cerebellum, microglial numbers increase slightly more than twice, going from postnatal days 5–10 [44]. In our study, cell culture was prepared from 5–7-day-old rat cerebella and monitored for 7 days in culture. Initially after seeding, the cell behavior might have been influenced by the isolation stress. From DIV2 to DIV7, microglial cell numbers in mixed neuronal-glial cultures increased about two times on the poly-L-lysine-coated plastic, about five times on the PEG-CLP hydrogel, and about seven times on the PEG-CLP-RGD hydrogel. Thus, the microglial proliferation rate on hydrogels was much higher than that observed in cerebellar tissue. However, it has to be taken into account that the cultures were grown with a fetal bovine serum supplement in the culture medium, which is known as microglia activating and proliferation stimulating factor [45]. Thus, microglial proliferation in all samples examined in this study might have been shifted to higher rates due to the effect of serum supplement.

The cytokine assessment revealed that such microglial proliferation was not associated with any acute inflammatory response; however, there was a slight increase in both TNFα and IL-1β levels of the PEG-CLP-RGD containing hydrogel samples. The increased numbers of glia in the developing cerebellum is a typical sign of intensive brain remodeling processes because the microglial cells are the primary responsible for the neuronal network formation by phagocytosing unnecessary neurons and pruning inefficient synapses [16,36]. Fetal microglia secrete pro-inflammatory cytokines including, IL-1, IL-6, and TNFα, which are important regulators of glial proliferation, developmental apoptosis, and synaptogenesis [46,47]. Moreover, both in development and during the whole life span, TNFα and IL-1β have been implicated in regulating synaptic transmission and functional plasticity in normal healthy brains [48]. The nanomolar concentrations of these cytokines are cytoprotective, but micromolar levels are neurotoxic and indicate neuroinflammation [49]. In our study, the increase of TNFα and IL-1β in the medium of cells grown on PEG-CLP and PEG-CLP-RGD hydrogels was even lower than the nanomolar level; thus, it was unlikely to be a sign of acute neuroinflammation. If the cultures were not in the inflammatory state, the next possible explanation of such microglial proliferation is that the cells isolated from the developing cerebella continued to actively participate in functional neuronal network formation. Altogether, these features allowed us to assume that the PEG-CLP and PEG-CLP-RGD substrates promoted a glial-neuronal communication characteristic for the brain developmental processes.

The important finding of the study is that the neuronal clusters on the PEG-CLP and PEG-CLP-RGD hydrogel membranes revealed the signs of spontaneous neuronal activity, as reflected in fast-rising Ca^2+^ signals on day 4 in culture. Though PEG-CLP induced faster neuronal maturation with a higher firing frequency and amplitude, the incorporation of the RGD motif ensured more qualitative synaptic signaling occurring later in time, characterized by a faster decay of intracellular Ca^2+^. Furthermore, the initial qualities of the PEG-CLP hydrogel-supported cultures become not significant later in time when the cultures reached DIV7. The pro-functional stimulation of the RGD peptide was also supported by its neurite outgrowth promoting activity that was so clearly visible in the neuritogenesis evaluation experiments. Such a fast and qualitative maturation of the functional cerebellar organoids suggests PEG-CLP and, especially, PEG-CLP-RGD as promising matrices for brain pathology in vitro modelling, as well as brain development, neuronal-glial communication, and other studies. Note that despite so many neurons in the cultures studied, there were no signs of functional activity on both glass and plastic poly-L-lysine-covered surfaces up to DIV7 in culture. Thus, to ensure functional cerebellar organoids, it is necessary to design matrices that not only ensure neuronal cell differentiation and viability but also promote enough glial cells for the neuronal maintenance and enough migratory capacity that would enable neurons to cluster and glial cells to respond to the neuronal signaling.

Regarding the current state of hydrogel use for brain in vitro cell research, the most popular hydrogels, according to the number of publications, are the cell culture-derived matrices Geltrex^TM^ and Matrigel^®^. They are used to develop human brain organoids from iPSCs and even applied for neural tissue engineering in animal studies [50,51,52]. The hydrogels well-support (although not so fast) neural cell differentiation and functional maturation [53,54,55]. Though providing a nice in vivo-like environment for neural cells due to the rich composition of basal lamina, such matrices of biological origin are not defined, are expensive, and slightly vary from batch to batch. Thus, such scaffolds are not very favorable for highly standardized automated high throughput screening applications and tissue regeneration. RADA peptide-based synthetic hydrogel PuraMatrix^TM^ functionalized with laminin also supports neural cell survival; however, to our knowledge, there are no functional brain cell culture models described on this scaffold. All the above mentioned and other commercially available hydrogel systems provide scientists with the opportunity to self-tune hydrogel composition and stiffness, opening a lot of options for experimentation. On the other hand, this raises the issue of the experimental standardization and comparison of the results produced by different researchers. The chemically crosslinked PEG-peptide hydrogel membranes examined in this study have robust and well-standardized structure and stiffness. Therefore, they might be well-suited for such neural cell culture applications that require a high level of reproducibility, such as drug screening or tissue engineering. 

## 5. Conclusions

In conclusion, we have shown that chemically crosslinked, synthetic peptide hydrogels composed of specific ECM protein mimicking fragments are advantageous for neural cell cultures. They provide a well-defined, tissue-like elastomechanical environment, and their biochemical composition can be tailored by choosing appropriate synthetic peptides. Second, we have shown that such ECM-mimetic matrices promote in the vivo-relevant organization, composition, morphology, and function of primary cerebellar cell cultures. The demonstrated level of in vitro organization cannot be achieved on a plane or poly-L-lysine-coated plastic or glass substrates, and, to the best of our knowledge, this fast functional maturation of self-assembled cerebellar organoids has not been reported on other hydrogel materials. Finally, the studied all-synthetic, shapeable, and optically transparent materials that ensure high cellular viability without inducing inflammatory responses are highly promising tools for developing advanced in vitro research systems, tissue-on-a-chip devices, and neural regenerative strategies.

## 6. Patents

The PEG-CLP hydrogel matrix technology described in this manuscript is disclosed in Ferentis UAB patents US102US10273287B2 and B2540116 and patent applications EP3283510A1, WO2016/165788A1.

## Figures and Tables

**Figure 1 biomolecules-10-00754-f001:**
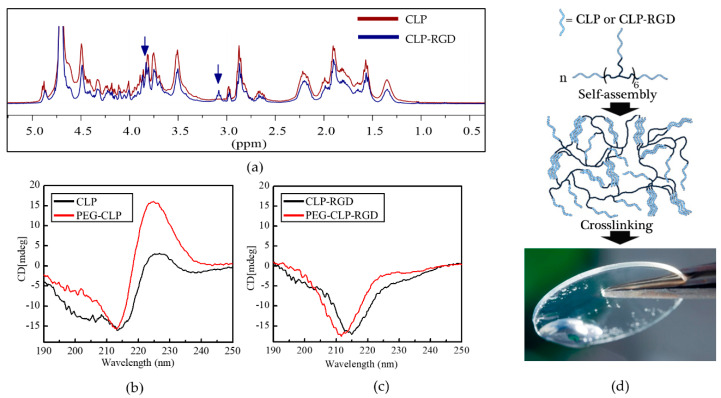
Spectroscopic characterization of the synthesized collagen-like peptide (CLP), CLP-integrin-binding motif arginine-glycine-aspartate (RGD), and the respective polyethylene glycol (PEG)-peptide conjugates. (**a**) NMR spectra of the peptides; blue arrows indicate arginine (δ3.19), and serine (δ = 3.83) peaks from CLP-RGD. (**b**) Circular dichroism (CD) spectra of the CLP peptide and the same peptide conjugated to 8-arm-PEG. (**c**) CD spectra of the CLP-RGD peptide and the same peptide conjugated to 8-arm-PEG. (**d**) Schematic explaining the principle PEG-peptide hydrogel synthesis. The peptides first were conjugated to a multi-arm PEG template, which facilitated their self-assembly in a solution. The formed architectures were then locked by further crosslinking the conjugates to obtain a self-supporting and optically transparent material.

**Figure 2 biomolecules-10-00754-f002:**
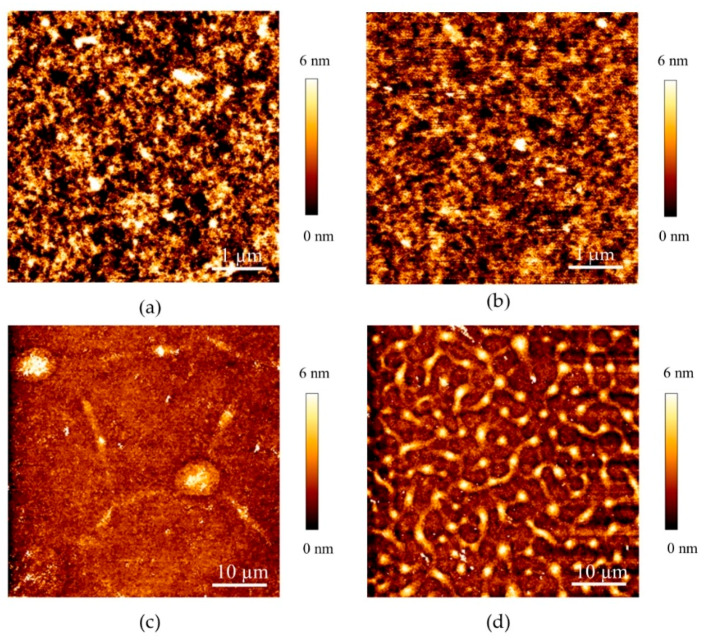
Atomic force microscopy topography images of peptide-PEG hydrogels and peptide-PEG coated glass surfaces. Surface topography of PEG-CLP (**a**), PEG-CLP-RGD (**b**) hydrogels and glass surfaces coated with the PEG-CLP (**c**) and PEG-CLP-RGD (**d**) conjugates. Images were acquired in PBS pH = 7.4 buffer solution.

**Figure 3 biomolecules-10-00754-f003:**
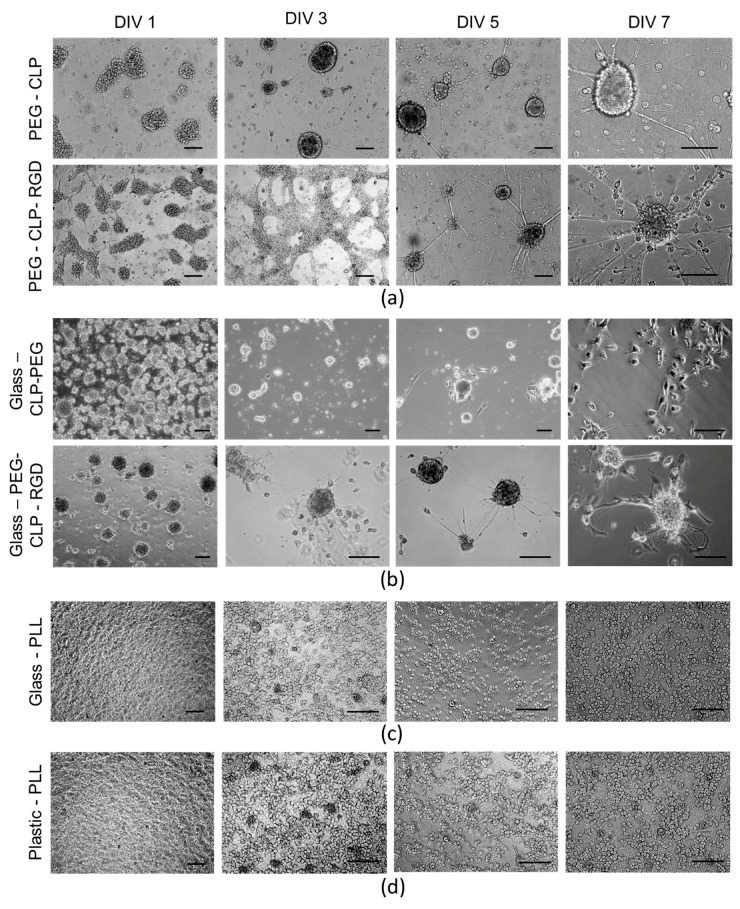
Phase-contrast images of alive cerebellar neuronal-glial cell cultures developing on PEG-CLP and PEG-CLP-RGD hydrogel membranes (**a**), on glass coated with PEG-CLP or PEG-CLP-RGD conjugates (**b**), on poly-L-lysine-coated glass (**c**), and on plastic (**d**). The small round cell bodies belong to granule neurons and the larger bodies belong to microglial cells. The astrocyte bodies are not visible enough in phase-contrast images. Scale bar is 100 µm.

**Figure 4 biomolecules-10-00754-f004:**
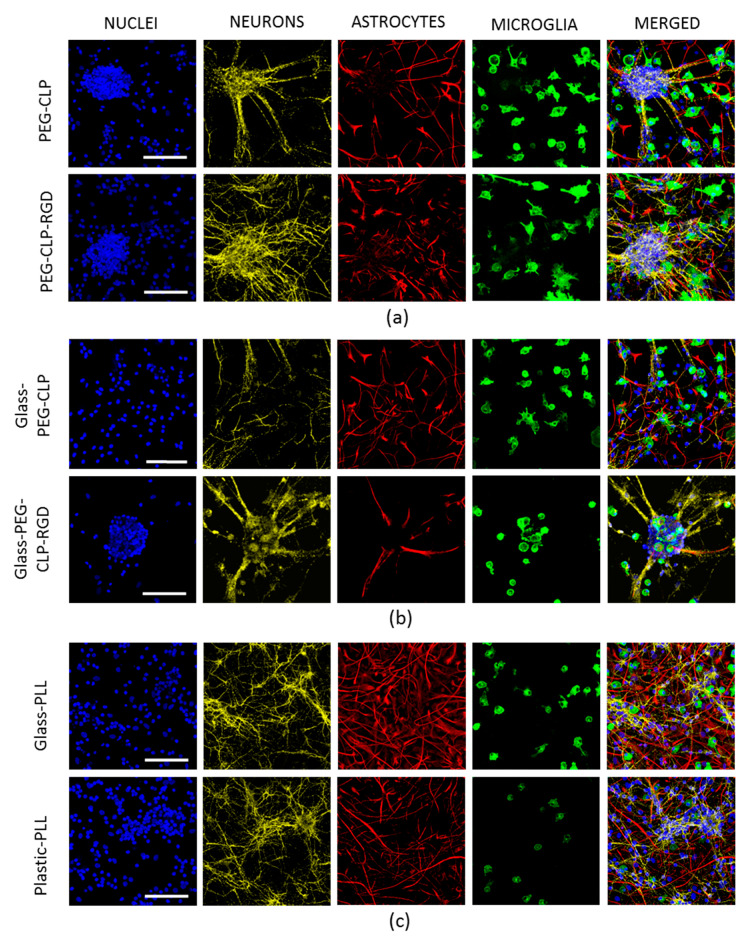
Confocal images of cerebellar neuronal-glial cell cultures after seven days in vitro (DIV7) on PEG-CLP and PEG-CLP-RGD hydrogel membranes (**a**), on glass coated with PEG-CLP and PEG-CLP-RGD conjugates (**b**), and on poly-L-lysine-coated glass and plastic (**c**). All nuclei were stained blue with Hoechst33342; neurons (yellow) are immunolabelled for microtubule-associated protein 2 (MAP-2), astrocytes (red) are immunolabelled for glial fibrillary acidic protein (GFAP), and microglia are stained green with isolectin GS-IB4. Scale bar is 100 μm.

**Figure 5 biomolecules-10-00754-f005:**
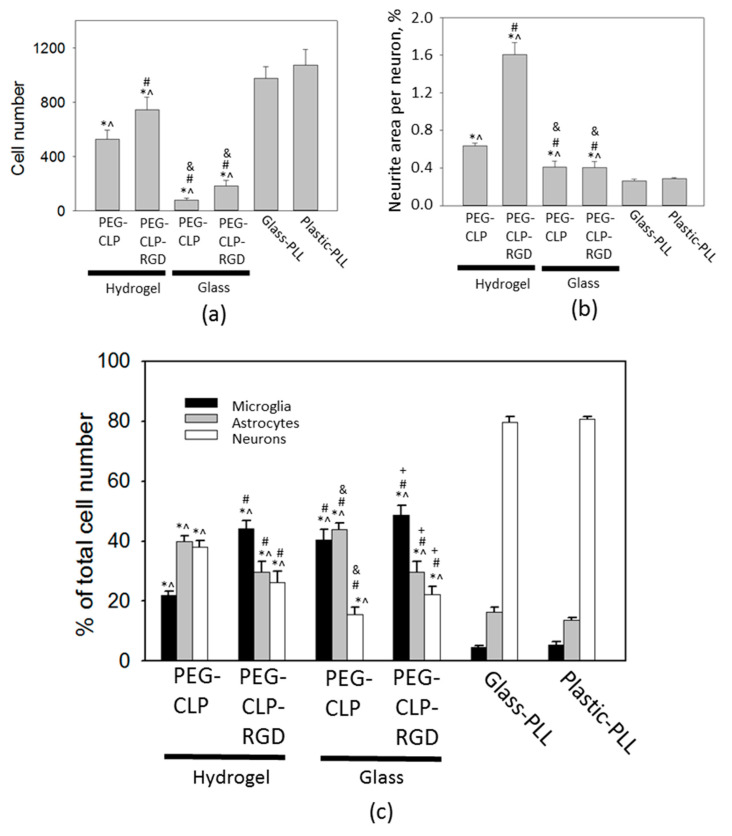
Quantitative comparison of the total cell number (**a**), neuritogenesis (**b**), and cellular composition (**c**) in cerebellar cell cultures on PEG-CLP and PEG-CLP-RGD hydrogels (membranes), on PEG-CLP- and PEG-CLP-RGD-coated glass, and on poly-L-lysine (PLL)-coated tissue culture glass and plastic substrates. Cell number in (**a**) was evaluated according to the number of Hoechst33342-positive nuclei in the 400 × 500 μm^2^ area of the confocal micrographs. The amount of each cell type in (**b**) is represented as the percentage of the total cell number in the cultures. For neurite evaluation, the anti-MAP2-positive area in micrographs was calculated by the ImageJ software and represented as the percentage of the total image area per neuronal nucleus. The data are shown as averages of 3–6 experiments with standard error. *—statistically significant difference compared to Glass-PLL; ^—statistically significant difference compared to Plastic-PLL; #—statistically significant difference compared to PEG-CLP hydrogel; &—statistically significant difference compared to PEG-CLP-RGD hydrogel; and +—statistically significant difference compared to CLP glass; *p* ≤ 0.01.

**Figure 6 biomolecules-10-00754-f006:**
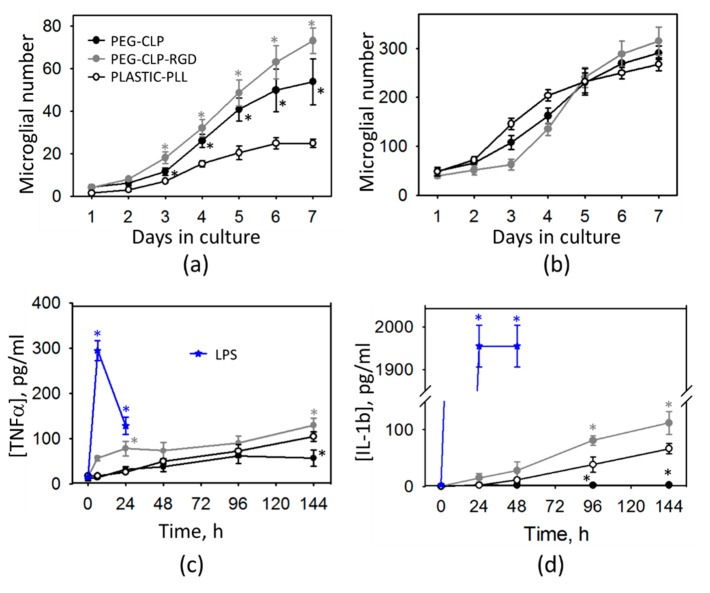
Increase in the microglial number on CLP-PEG and CLP-RGD-PEG hydrogels and poly-L-lysine (PLL)-covered plastic per microscopic image in mixed neuronal-glial (**a**) and pure microglial cultures (**b**); the amount of inflammatory cytokines TNF-α (**c**) and IL-1β (**d**) in the medium of neuronal-glial cells on both hydrogels and plastic. LPS (lipopolysaccharide)—cells on the PEG-CLP hydrogel were activated by adding 100 ng/mL lipopolysaccharide for the positive control of cytokine release. The data are presented as averages of 3–5 experiments with standard error. Statistical analysis: one way ANOVA; *—statistically significant difference compared to Plastic-PLL; *p* ≤ 0.01.

**Figure 7 biomolecules-10-00754-f007:**
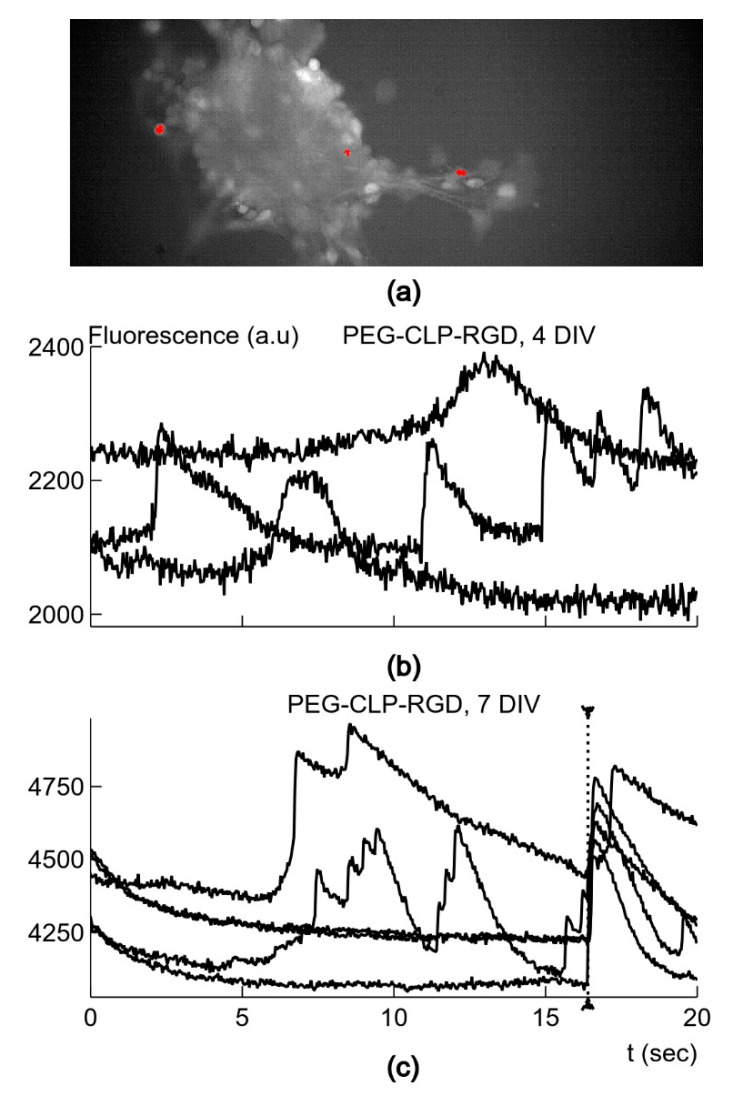
(**a**) Fluorescence image of culture after 4 days on a PEG-CLP-RGD hydrogel membrane with spontaneously active neurons (in red). (**b**) Traces of spontaneous fluorescence changes from 3 different neurons marked in (**a**) and recorded simultaneously. (**c**) Fluorescence traces from 5 different neurons after DIV7 on a PEG-CLP-RGD hydrogel membrane; dotted line marks apparent synchronization between neurons. In (**b**), the time axis is the same as in (**c**).

**Figure 8 biomolecules-10-00754-f008:**
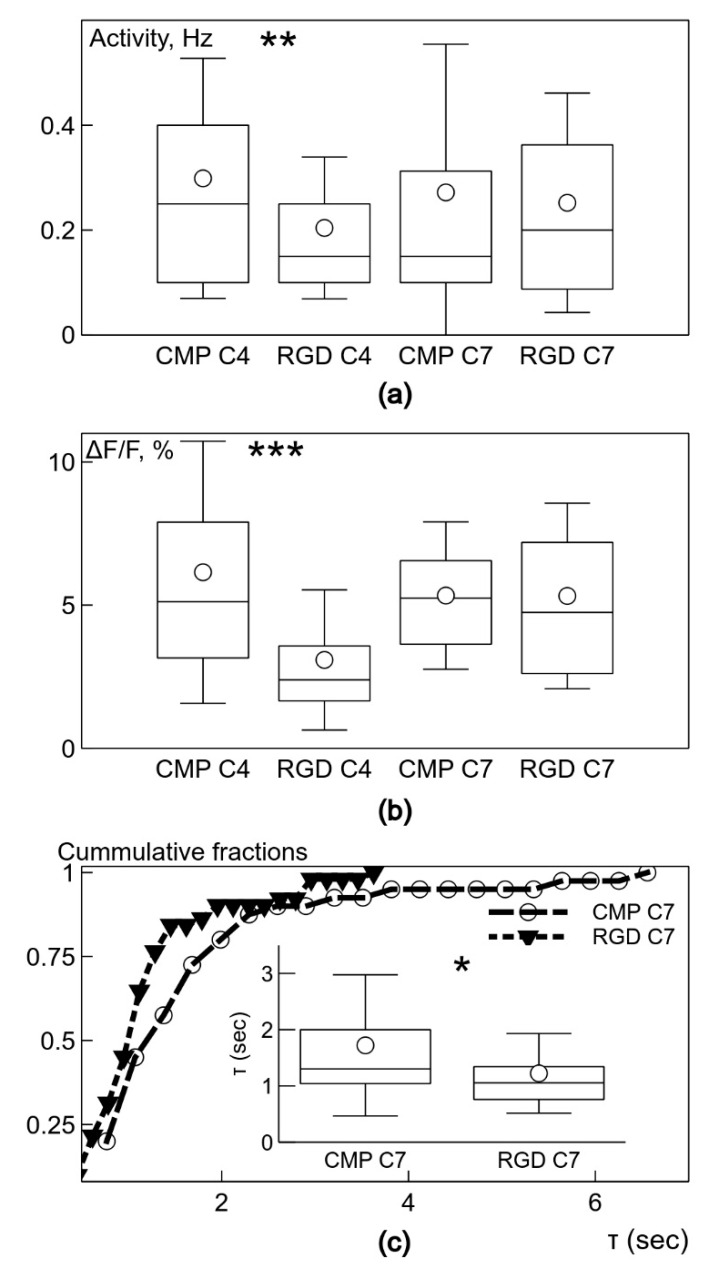
Box-and-whisker plots showing the frequency of the spontaneous activity, Hz, (**a**) and relative signal strength, δF/F (**b**) of neurons grown after 4 and 7 days in vitro in cultures on PEG-CLP or PEG-CLP-RGD hydrogels (indicated under abscissa). (**c**) The decay of the spontaneous signals after 7 days, as illustrated by the characteristic constant, τ, having values shown as both cumulative probability distributions and box-and-whisker plots (inset). DIV indicates day in vitro. Error bars indicate 1 standard deviation, the shoulders of boxes indicate the 25–75% intervals, the median of data is highlighted by a horizontal line, and the mean is indicated by a circle. * *p* < 0.05, ** *p* < 0.01, *** *p* < 0.001 (Student *t*-test).

**Table 1 biomolecules-10-00754-t001:** Elastomechanical properties of crosslinked peptide hydrogels. Shear storage modulus (G’) was estimated by rheology and was used to calculate the Young (elastic) modulus (E). Direct and detailed probing of the Young moduli (E*) was performed by the atomic force microscopy nanoindentation technique, and the values for PEG-CLP compared to PEG-CLP-RGD had a statistically significant difference, *p* ≤ 0.05.

Hydrogel	G’, kPa	E, kPa	E*, kPa
PEG-CLP	26.7 ± 2.1	80.1 ± 6.3	144.5 ± 26.5
PEG-CLP-RGD	20.4 ± 5.5	61.2 ± 16.5	75.6 ± 19.0

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
