# Peer review of "Cerebellar Cells Self-Assemble into Functional Organoids on Synthetic, Chemically Crosslinked ECM-Mimicking Peptide Hydrogels"

_biomolecules, 2020, doi:10.3390/biom10050754_

Round 1

Reviewer 1 Report

Summary:

This manuscript describes the evaluation of hydrogels consisting of poly(ethylene glycol) and collagen-like peptides (CLPs) with or without RGD, an integrin binding domain, for supporting the development of cerebellar organoids. The authors demonstrate the ability to culture multiple neuronal cell types and form aggregates that develop neurites and show calcium oscillations that more closely mimic in vivo behavior than cells cultured on functionalized glass or plastic. Overall, I believe that this paper would be of great interest to readers; however, there are some issues that need to be addressed before I can recommend publication. Chief among these points are (1) providing additional data to demonstrate the synchronization of neurons, which is currently very lacking, and (2) either a new experimental control group with a scrambled RGD sequence (e.g. GRD) to match many of the non-focal adhesion impact of adding that sequence, such as altered supramolecular assembly or at the very least, an enhanced discussion on the impact of including this sequence.

Major Points:

  1. Can the authors please comment on why they used PEG-CLP rather than PEG-CLP-RDG (or some other scramble? It seems as if the addition of RGD does have some impact on the properties, such as E*, G’, and E, possibly due to the interference with proper triple helix formation. Can they also please note if these differences were statistically significant?
  2. The paper refers to supplementary figures, but I did not have access to them.
  3. Figure 4 is missing.
  4. It would be very helpful to talk about what the desirable outcomes are for cell culture. For example, is there a certain percentage of microglia that are optimal? How proliferative do you want them to be? How much neurite area is best per neuron? How does these compare to the in vivo cerebellar microenvironment?
  5. Synchronized activity is touted as an especially exciting result, however, I do not see evidence of this type of activity. The only data presented that would even be able to show this is Figure 7B, but it does not appear to be synchronized at all. Is there other data that could be presented to show this effect?

Minor Points:

  1. Does the AFM used really have accuracy down to 0.01 nm (i.e. 0.1 angstrom)? That is smaller than an atom. If not, please reduce the significant digits used to report these values.
  2. Figure 2 would be easier to interpret if a uniform color scale (0-10 nm) were used.
  3. Figure 3C should be labeled Figure 3C and 3D.
  4. What does “neuritogenesis was extremely intensive reaching 2% of total cell culture area covered by neurites per one neuronal cell” (lines 455-456) mean? Please clarify.
  5. Can the authors please comment on the seemingly high levels of TNF generated in the hydrogel cultures? It is true that they are not as high as upon LPS stimulation, but LPS is a very potent immunostimulant that induces a large inflammatory response. Does this amount fall within the desirable range (lines 621-623)?
  6. In lines 546-547, please qualify the statement “However, the difference was not significant due to the small number of the performed measurements” since this is not a certainty.

Author Response

Response to the comments of the Reviewer 1

Reviewer 1 – general comment:

This manuscript describes the evaluation of hydrogels consisting of poly(ethylene glycol) and collagen-like peptides (CLPs) with or without RGD, an integrin binding domain, for supporting the development of cerebellar organoids. The authors demonstrate the ability to culture multiple neuronal cell types and form aggregates that develop neurites and show calcium oscillations that more closely mimic in vivo behavior than cells cultured on functionalized glass or plastic. Overall, I believe that this paper would be of great interest to readers; however, there are some issues that need to be addressed before I can recommend publication.

Response: We thank the Reviewer for the careful reading of the MS and addressing not only several important aspects of our work, but also suggesting interesting topics for our upcoming studies. We do appreciate a number of suggestions for improving the clarity and strengthening the experimental evidences, we hope we have taken them into account properly as explained below. Also, we thank for noticing and reporting a number of technical issues that occurred during the original submission of the MS.

Chief among these points are (1) providing additional data to demonstrate the synchronization of neurons, which is currently very lacking

Response: We thank the Reviewer for pointing out that the comments about neuronal synchronization were not supported by the data presented in the MS. We have carefully revised both the data and text – see respond to the Major point 5.

…and (2) either a new experimental control group with a scrambled RGD sequence (e.g. GRD) to match many of the non-focal adhesion impact of adding that sequence, such as altered supramolecular assembly or at the very least, an enhanced discussion on the impact of including this sequence.

Response: We appreciate Reviewer’s suggestion on paying more attention to the mechanisms behind the interaction of the cells and synthesized material (peptides and peptide-based hydrogels). However, we do not fully agree with the recommendation to elaborate in more detail on the possible alternative effects of the RGD sequence in the present paper, at least not by performing additional experiments with the scrambled RGD sequences.

First, we were able to show that the cells tend to adhere more on the RGD-containing surface compared to CLP without RGD independently of its elastomechanical properties (note that the elastomechanical response of such ultrathin films of both PEG-peptide conjugates is dominated by the glass substrate).

Second, the covalent attachment of the conjugates to the glutaraldehyde-terminated glass occurs via the peptides themselves (via their amine groups). While we have not performed any studies on the actual attachment stoichiometry and composition (how many amines per a (multi-arm) PEG-peptide conjugate pin to the surface and how many peptide molecules of the same conjugate are available for interaction with the cell surface), one normally expects less freedom for forming supramolecular assemblies in this kind of coatings as compared to solution/hydrogel. This is due the multiple and random pinning to the surface, as well as due to the subsequent handling (rinsing, storing in the dry state) of the coated glass samples.

Third, RGD sequence is reported to have opposite but not similar bioactivity on brain cell proliferation, adhesion, motility, survival and differentiation compared to the scrambled version of the peptide [1,2].

Taken together the above presented evidence suggests that the cells more likely recognize the presence of the RGD sequence directly, than react to nonspecific peptide signaling.

Nevertheless, we completely agree with the Reviewer 1 that it would be important to experimentally investigate the impact of adhesion mechanism on cell culture formation and neuronal functional maturation on PEG-CLP and PEG-CLP-RGD hydrogels in the future studies, and we definitely will take into account the suggestion to introduce scrambled peptide as control in this research. To obtain a detailed view of the actual interaction mechanisms, one should not only follow the recommendation of the Reviewer to synthesize hydrogels containing scrambled RGD sequences. In fact, one would need to design an extended study on synthesis and advanced characterization of a series of respective model surfaces, where the surface density, orientation, lateral interactions, and steric accessibility of the respective peptide molecules on solid substrates is fully controlled. And then to employ such hydrogels and model surfaces for dissection and probing any direct and indirect interactions with the adhesion machinery of the selected cell types. Having our background in surface science we know that such a study would be both interesting and feasible, yet a time consuming and technically demanding project. The aim of the current study was to compare (1) how CLP and CLP-RGD peptides influence cell behavior and function, and (2) how this peptide influence is affected by mechanical properties of the substrate the peptides are attached to. Thus, investigation of the focal/non-focal adhesion formation by introducing scrambled RGD peptide would be certainly well beyond the scope of the present paper. Nevertheless, we agree with the Reviewer’s suggestion to enhance the discussion, which obviously indeed was not structured in the best way.

Actions taken:

  • We have introduced discussion about the possible interactions between the tethered peptides and cells in Discussion section (L602-621).
  • We have included a clear statement in the same paragraph that the respective PEG-peptide conjugates are covalently bound to the glass surface randomly via the peptide molecules, whereas the remaining unbound peptide blocks sitting on the PEG templates would have limited availability for forming supramolecular complexes on the surface. Thus, the RGD motif itself obviously promotes cellular attachment to the surface.
  • We have provided additional references and discussion on the effects of RGD in similar cell/tissue systems.

Major Points:

  1. Can the authors please comment on why they used PEG-CLP rather than PEG-CLP-RDG (or some other scramble? It seems as if the addition of RGD does have some impact on the properties, such as E*, G’, and E, possibly due to the interference with proper triple helix formation. Can they also please note if these differences were statistically significant?

Response: The overall idea was to show the advantage of the synthetic approach in modelling the cellular environment of the native cerebellum. Collagen like peptide, or CLP, was used as synthetic analogue of collagen, and RGD was used to imitate the signaling of fibronectin.

At the same time, from the practical point of view of hydrogel synthesis, we were testing our concept of biomimetic ECM modeling. Instead (or beside) of synthesizing the hydrogel composed from different peptides (e.g. CLP and RGD as separate blocks), we suggest employing “chimeric”, multifunctional peptides. Our present study must be seen only as the first attempt into this direction. Our ambition is to perform next studies where the biomimetic ECM is synthesized from more different peptide sequences and peptide compositions, also including additional functional post-modifications of hydrogels. One should not also exclude a combinatorial approach of synthesizing and comparing many different peptide scrambles and hydrogel compositions for specific tissue modeling/regeneration purposes. At the present stage, our study demonstrates the feasibility of such strategies towards the new generation of biomaterials.

More specifically to the present study, while we have seen the effect of the RGD sequence on both the conformation of the core CLP molecule and elastomechanics of the hydrogel, we would like to draw the attention of the Reviewer that both hydrogel compositions (PEG-CLP and PEG-CLP-RGD) fall in the stiffness range of brain tissues (See Ref 42 in the manuscript).

Also, we would like to stress that despite the very initial characterization provided in the present MS, the actual spatial structures of the CLP and CLP-RGD blocks in the hydrogels remain unknown. Resolving them unambiguously would require advanced structural and theoretical studies. Only having those data, one could depart to more systematic studies on the conformational and supramolecular effects on the cell-surface and cell-cell interactions.

Action taken:

  • We have added a sentence in the Introduction section (L77 in the revised MS) on the rationale behind using chimeric peptides such as CLP-RGD.
  • The Young moduli (E*) values directly probed for PEG-CLP compared to PEG-CLP-RGD hydrogel have a statically significant difference, p≤0.05. We have now included this information in Table 1.

  1. The paper refers to supplementary figures, but I did not have access to them.

Response: We are really sorry for this technical issue. The supplementary figure was probably lost during/after the uploading. We will double check to ensure this is added.

  1. Figure 4 is missing.

Response: Again, we are sorry for the inconvenience. We have seen the figure in the MS after uploading on the submission system, it is also inside the version the Editor have emailed us to check the text similarities with other papers. Thus, we have no idea why the Figure is lacking in the version generated for revision. We will definitely point it out to the Editor and hopefully the issue will be solved.

  1. It would be very helpful to talk about what the desirable outcomes are for cell culture. For example, is there a certain percentage of microglia that are optimal? How proliferative do you want them to be? How much neurite area is best per neuron? How does these compare to the in vivo cerebellar microenvironment?

Response: Although increased microglial proliferation and amoeboid morphology of the cells is often attributed to the signs of inflammation, such features are also characteristic to the developing brain. Perez-Pouchoulen and colleagues have found that microglial percentages in developing rat cerebellum vary from about 20 to 60 during postnatal 21 day period [3]. Microglial numbers on both PEG-CLP and PEG-CLP-RGD hydrogels and glass surfaces were within this range indicating the materials support in vivo-relevant microglial/non-microglial cell ratios. On the other hand, the total cell numbers on PEG-peptide-coated glass surfaces were more than 5 times lower than on hydrogels and did not support functional cell assemblies. On poly-L-lysine-coated plastic or glass, microglial percentages were far below those found in real cerebellar tissue. In rat cerebellum, microglial numbers increase slightly more than twice going from postnatal day 5 to day 10 [3]. In our study, cell culture was prepared from 5-7 day old rat cerebella and monitored 7 days in culture. Initially after seeding, the cell behavior might be influenced by the isolation stress. From day in vitro 2 to day in vitro 7, microglial cell numbers in mixed neuronal-glial cultures increased about 2 times on poly-L-lysine-coated plastic, about 5 times on PEG-CLP hydrogel and about 7 times on PEG-CLP-RGD hydrogel. Thus, microglial proliferation rate on hydrogels was much higher than that observed in cerebellar tissue. However, it has to be taken into account that the cultures were grown with fetal bovine serum supplement in the culture medium which is known as microglia activating and proliferation stimulating factor [4]. Thus, microglial proliferation in all samples examined in this study might be shifted to higher rates due to the effect of serum supplement.

Regarding optimal neurite area per neuron, we assume that the larger area covered by the neurites of each neuron corresponds to the better ability of the culture to form functional neuronal network. Although it would be possible to calculate the exact area per neuron in mm and try to compare to the length of neurites of cerebellar granule neuron in tissue, such evaluation would be very inaccurate due to area enlargements in places of brighter fluorescence, and also because the area covered by neuronal bodies that are also stained MAP-2-positive. Therefore, we find such evaluation is only correct for comparison between different samples of the same study but not for extrapolation to in vivo situation and comparison to the real length of neurites of the cerebellar granule neurons.

Action taken: The discussion about in vivo-relevance of microglial number and proliferation data is now introduced in the Discussion section (L641-658).

  1. Synchronized activity is touted as an especially exciting result, however, I do not see evidence of this type of activity. The only data presented that would even be able to show this is Figure 7B, but it does not appear to be synchronized at all. Is there other data that could be presented to show this effect?

Response: In our data synchronization appeared in about half of recordings in C7 cultures, but the data obtained was not large enough to build cross-correlograms in order to prove significance of synchronization between spontaneously active neurons. We changed text in the Abstract:

“Both compositions promoted spontaneous organization of primary cerebellar cells to tissue-like clusters with fast rising Ca2+ signals in soma reflecting action potential generation. Notably, neurons on PEG-CLP-RGD had more neurites and better synaptic efficiency compared to PEG-CLP. For comparison, Poly-L-Lysine-coated glass and plastic surfaces did not induce formation of such spontaneously active networks.”

  • we changed 1st paragraph in the Introduction on line 61 [p. 2]:

“Referring specifically to the brain cells, the challenge lies in providing a robust and repeatable assay ensuring tissue-like cell composition and interaction as well as efficient maturation of the functional network characterized by spontaneous neuronal firing activity.”

  • we changed the last paragraph in the Discussion on line 677 [p. 27, the 2nd para] :

The important finding of the study is that the neuronal clusters on the PEG-CLP and PEG-CLP-RGD substrates revealed the signs of spontaneous neuronal activity reflected in fast-rising Ca2+ signals on day 4 in culture.

 Minor Points:

  1. Does the AFM used really have accuracy down to 0.01 nm (i.e. 0.1 angstrom)? That is smaller than an atom. If not, please reduce the significant digits used to report these values.

Response: The Reviewer obviously refers to the ellipsometry data in L351, P9 in the original MS, as the AFM data is provided at an accuracy of 1 Å throughout the text. Regarding the provided ellipsometric data, we would like to note that this method is based on modeling the increments at the different steps of surface synthesis/binding process as efficient interfacial layers. The theoretical sensitivity to thickness in such an approach is on the order of 0.01nm and such an accuracy is normally used in representing the ellipsometric data in the literature. For more details on precision of the ellipsometry technique see for example https://arxiv.org/ftp/arxiv/papers/1210/1210.1076.pdf

Action taken: We have not changed the significant digits, however in the Methods section we now specify that all the thickness and roughness values are provided as mean values ±SDVs.

  1. Figure 2 would be easier to interpret if a uniform color scale (0-10 nm) were used.

Action taken: We have applied the same color scales of 0-6 nm in Figure 2 to enhance the contrast of the AFM images and for easier comparison of the different surfaces.

  1. Figure 3C should be labeled Figure 3C and 3D.

Action taken: Panel 3 (d) is introduced to the Figure 3 as suggested.

  1. What does “neuritogenesis was extremely intensive reaching 2% of total cell culture area covered by neurites per one neuronal cell” (lines 455-456) mean? Please clarify.

Response: According to the assay of neuritogenesis evaluation used in the study, the total area of MAP-2-indicating fluorescence in the micrographs was divided for the total number of neuronal nuclei resulting in the average neurite area for one neuron. In PEG-CLP-RGD cultures, this parameter was much higher compared to other samples. However, as explained above in response to point 4, such evaluation of neuritogenesis allows only to compare the level between different samples of the same (or other the same way conducted) study.

  1. Can the authors please comment on the seemingly high levels of TNF generated in the hydrogel cultures? It is true that they are not as high as upon LPS stimulation, but LPS is a very potent immunostimulant that induces a large inflammatory response. Does this amount fall within the desirable range (lines 621-623)?

Response: Indeed, fetal microglia are known to secrete pro-inflamamtory cytokines including IL-1 and TNFa, which are important regulators of glial proliferation, developmental apoptosis and synaptogenesis [6,7]. Moreover, both in development and during the whole life span, TNFα and IL-1β have been implicated in regulating synaptic transmission and functional plasticity in normal healthy brain [8]. Nanomolar concetrations of these cytokines are cytoprotective, but micromolar levels are neurotoxic and indicate neuroinflammation [9]. In our study, the increase of TNFα and IL-1β in the medium of cells grown on PEG-CLP and PEG-CLP-RGD hydrogels is even lower than nanomolar level, thus, it is unlikely to be a sign of acute neuroinflammation but rather of more active involvement of microglia in neuronal selection and synaptic pruning.

Action taken: We have expanded the discussion about the inflammatory cytokines in the Discussion section accordingly.

  1. In lines 546-547, please qualify the statement “However, the difference was not significant due to the small number of the performed measurements” since this is not a certainty.

Response: We wanted point out that the decay of the signals in neurons on day 4 in vitro on the poly-L-lysine-coated tissue culture plastic was much slower, τ was equal to 2.26±1.28 s comparing to 1.4s on hydrogel cultures. The difference was not significant because neurons grown poly-L-lysine-coated plastic had very low probability to generate spontaneous Ca2+ signals, and the total number of spontaneous signal cases found on plastic were not enough for statistical analysis. We removed the following text on line 653 in previous version of the manuscript [p. 24, 3rd paragraph]:

“The decay of the signals in neurons on day 4 in vitro on the poly-L-lysine-coated tissue culture plastic was much slower, τ was equal to 2.26±1.28 s (n=8). However, the difference was not significant due to the small number of the performed measurements.”

References

  1. Mattern, R.-H.; Read, S. B.; Pierschbacher, M. D.; Sze, C.-I.; Eliceiri, B. P.; Kruse, C. A. Glioma cell integrin expression and their interactions with integrin antagonists: Research Article. Cancer Ther. 2005, 3A, 325–340.
  2. Madl, C. M.; Katz, L. M.; Heilshorn, S. C. Bio-Orthogonally Crosslinked, Engineered Protein Hydrogels with Tunable Mechanics and Biochemistry for Cell Encapsulation. Adv. Funct. Mater. 2016, 26, 3612–3620, doi:10.1002/adfm.201505329.
  3. Perez-Pouchoulen, M.; VanRyzin, J. W.; McCarthy, M. M. Morphological and phagocytic profile of microglia in the developing rat cerebellum. eNeuro 2015, 2, doi:10.1523/ENEURO.0036-15.2015.
  4. Bohlen, C. J.; Bennett, F. C.; Tucker, A. F.; Collins, H. Y.; Mulinyawe, S. B.; Barres, B. A. Diverse Requirements for Microglial Survival, Specification, and Function Revealed by Defined-Medium Cultures. Neuron 2017, 94, 759-773.e8, doi:10.1016/j.neuron.2017.04.043.
  5. Jorfi, M.; D’Avanzo, C.; Kim, D. Y.; Irimia, D. Three-Dimensional Models of the Human Brain Development and Diseases. Adv. Healthc. Mater. 2018, 7, 1700723, doi:10.1002/adhm.201700723.
  6. Giulian, D.; Young, D. G.; Woodward, J.; Brown, D. C.; Lachman, L. B. Interleukin-1 is an astroglial growth factor in the developing brain. J. Neurosci. 1988, 8, 709–714, doi:10.1523/jneurosci.08-02-00709.1988.
  7. Angeles Muñoz-Fernández, M.; Fresno, M. The role of tumour necrosis factor, interleukin 6, interferon-γ and inducible nitric oxide synthase in the development and pathology of the nervous system. Prog. Neurobiol. 1998, 56, 307–340.
  8. Werneburg, S.; Feinberg, P. A.; Johnson, K. M.; Schafer, D. P. A microglia-cytokine axis to modulate synaptic connectivity and function. Curr. Opin. Neurobiol. 2017, 47, 138–145.
  9. Estes, M. L.; McAllister, A. K. Alterations in Immune Cells and Mediators in the Brain: It’s Not Always Neuroinflammation! Brain Pathol. 2014, 24, 623–630, doi:10.1111/bpa.12198.

Reviewer 2 Report

The manuscript reported in vitro neural cell culture in two types of all-synthetic peptide hydrogels and on the same peptide coated glass substrate pairs. The authors have done a thorough characterization of the hydrogels and the coated glass substrate and did a great job in evaluation of cell culture in different substrates. The results suggested, both peptide hydrogens tested in the study show better supporting and allowed tissue-like clusters and synchronized networks formation comparing to glass substrates. Moreover, the introduction of RGD motif to CLP promotes cellular adhesion and enhances the cell migration and therefore more scattered cell distribution was observed. The manuscript was well-written, the experiments were well designed, and the results were well discussed. I would suggest publication after the authors address the following minor issues.

Page 1 Line30, please provide the full term of RGD in the abstract.

Figure 1. The figure resolution is low. Please change to a higher resolution image.

Table 1. Please include P values of the properties test in this table.

Figure 4 is not included in the manuscript. Please add the figure.

Author Response

Response to the comments of the Reviewer 2

The manuscript reported in vitro neural cell culture in two types of all-synthetic peptide hydrogels and on the same peptide coated glass substrate pairs. The authors have done a thorough characterization of the hydrogels and the coated glass substrate and did a great job in evaluation of cell culture in different substrates. The results suggested, both peptide hydrogens tested in the study show better supporting and allowed tissue-like clusters and synchronized networks formation comparing to glass substrates. Moreover, the introduction of RGD motif to CLP promotes cellular adhesion and enhances the cell migration and therefore more scattered cell distribution was observed. The manuscript was well-written, the experiments were well designed, and the results were well discussed. I would suggest publication after the authors address the following minor issues.

Response: We thank the Reviewer for taking the time to revise our study, for the positive attitude and improving comments.

Page 1 Line30, please provide the full term of RGD in the abstract.

Action taken: we include full amino acid names of RGD tripeptide in the abstract.

Figure 1. The figure resolution is low. Please change to a higher resolution image.

Action taken: Resolution of Figure 1 is now enhanced.

Table 1. Please include P values of the properties test in this table.

Action taken: The Young moduli (E*) of PEG-CLP compared to PEG-CLP-RGD have a statically significant difference, p≤0.05. We have now included p values in the Table 1.

Figure 4 is not included in the manuscript. Please add the figure.

Response: We are sorry for the inconvenience. We have seen the figure in the MS after uploading on the submission system, it is also inside the version the Editor have emailed us to check the text similarities with other papers. Thus, we have no idea why the Figure is lacking in the version generated for revision. We will definitely point it out to the Editor and hopefully the issue will be solved.

Round 2

Reviewer 1 Report

The authors have adequately addressed my concerns in their updated manuscript. Reducing claims about synchronization was key to not overstating their conclusions, as was showing new data in Figure 7c supporting the presence of this phenomenon, if only anecdotally. I do have a couple of minor suggestions regarding the new content added, however.

  1. It would be easier to compare the images in Figure 4, which is now visible in the paper, if the subpanels were at the same magnification. For some reason, they seem to vary by a factor of up to 3x (Glass-PEG-CLP-RGD vs Plastic-PLL), which makes it difficult to qualitatively compare.
  2. The text inserted on lines 602-621 needs to be revised to address fairly extensive grammatical issues.
  3. “Synchronized activity” should be removed from the keywords since this claim has been reduced.

Author Response

Revision Round 2

Response to the comments of the Reviewer 1

Reviewer 1: The authors have adequately addressed my concerns in their updated manuscript. Reducing claims about synchronization was key to not overstating their conclusions, as was showing new data in Figure 7c supporting the presence of this phenomenon, if only anecdotally. I do have a couple of minor suggestions regarding the new content added, however.

  1. It would be easier to compare the images in Figure 4, which is now visible in the paper, if the subpanels were at the same magnification. For some reason, they seem to vary by a factor of up to 3x (Glass-PEG-CLP-RGD vs Plastic-PLL), which makes it difficult to qualitatively compare.

Response: Thank you for the reasonable comment. We have now inserted all the subpanels of the Figure 4 at the same or very similar magnification.

  1. The text inserted on lines 602-621 needs to be revised to address fairly extensive grammatical issues.

Response: We have carefully revised the indicated text and corrected the grammatical mistakes.

  1. “Synchronized activity” should be removed from the keywords since this claim has been reduced.

Response: We have removed “synchronized activity” from the keywords, sorry for the carelessness.